# A Global Depth-Range-Free Multi-View Stereo Transformer Network with Pose Embedding

**Yitong Dong**[1][*]  **Yijin Li**[1][*]  **Zhaoyang Huang**[2]  **Weikang Bian**[2]
**Jingbo Liu**[1]  **Hujun Bao**[1]  **Zhaopeng Cui**[1]  **Hongsheng Li**[2]  **Guofeng Zhang**[1][†]
[1]State Key Lab of CAD&CG, Zhejiang University  [2]CUHK MMLab

## Abstract

In this paper, we propose a novel multi-view stereo (MVS) framework that gets rid of the depth range prior. Unlike recent prior-free MVS methods that work in a pair-wise manner, our method simultaneously considers all the source images. Specifically, we introduce a Multi-view Disparity Attention (MDA) module to aggregate long-range context information within and across multi-view images. Considering the asymmetry of the epipolar disparity flow, the key to our method lies in accurately modeling multi-view geometric constraints. We integrate pose embedding to encapsulate information such as multi-view camera poses, providing implicit geometric constraints for multi-view disparity feature fusion dominated by attention. Additionally, we construct corresponding hidden states for each source image due to significant differences in the observation quality of the same pixel in the reference frame across multiple source frames. We explicitly estimate the quality of the current pixel corresponding to sampled points on the epipolar line of the source image and dynamically update hidden states through the uncertainty estimation module. Extensive results on the DTU dataset and Tanks&Temple benchmark demonstrate the effectiveness of our method. The code is available at our project page: https://zju3dv.github.io/GD-PoseMVS/.

## 1 Introduction

Multi-view stereo matching (MVS) is a crucial technique in 3D reconstruction, which aims to recover robust and reliable 3D representations from multiple RGB images [1; 2; 3]. Traditional methods [4; 5; 6] rely on hand-crafted similarity metrics and regularizations to compute dense correspondences between the input images. These methods are prone to degradation in challenging scenarios, such as varying illumination, textureless regions, and occlusion regions. Recently, learning-based methods [7; 8; 9; 10; 11] directly learn discriminative features from the input images through neural networks such as CNN and Transformers. By sampling some possible depth hypothesis within a given depth range, they warp the features from the source images to the reference view (i.e., the plane sweep algorithm [12]) and compute the cost volume, which is then regularized also through the neural network to obtain the final depth maps. However, obtaining a suitable depth range is non-trivial when applied in real-world scenarios while these methods are generally sensitive to the depth range, which limits their application.

To get rid of the dependence on depth range, some methods [13; 14; 15] transform the regression problem in the given depth space into a matching problem on the epipolar lines. Similar to optical flow [16] and feature matching [17; 18; 19; 20], these methods also adopt a pair-wise manner. For example, DispMVS [14] computes the depth map of the source image multiple times through pairs

---

[*]Equal contribution

[†]Corresponding author

38th Conference on Neural Information Processing Systems (NeurIPS 2024).

that contain different source images and computes the final depth map by weight of sum. However, the pair-wise manner neglects the inter-image correspondence between the source images and could lead to sub-optimal solutions. Meanwhile, although DispMVS mitigates the influence of depth priors on constructing the 3D cost volume, its initialization based on depth range can still lead to significant performance degradation when the depth range error is too large, as shown in Fig. 1.

We argue that these methods need to consider all the source images at the same time. Our ideas are inspired by the recent methods [21; 22] of optical flow which concurrently estimate optical flows for multiple frames by sufficiently exploiting temporal cues. However, we find these frameworks cannot be trivially applied in the task of multi-view stereo. The reasons are twofold. First, a strong cue in the multi-frame optical flow estimation is that the flow originating from the same pixel belongs to a continuous trajectory in the temporal dimension. Additionally, the frames are sequentially aligned along this temporal dimension. Such inductive bias makes it easy to learn. But in the context of multi-view stereo, the source images may be captured in no particular order, lacking a similar constraint of continuity. Unlike optical flow, the input images in multi-view stereo are unordered. These distinctions pose a significant challenge when attempting to adapt the multi-frame optical flow framework for use in multi-view stereo. Second, the arbitrary positions and viewing angles of the source images, coupled with potentially large temporal gaps between captures, exacerbate issues such as varying illumination, significant viewport differences, and occlusions which call for new designs.

Based on the above observations, in this paper, we propose a novel framework that gets rid of the depth range assumption. Unlike some recent methods [13; 14; 15] that work in a pair-wise manner, the proposed method estimates the depth maps of a reference image by simultaneously considering all the source images. To address the first issue, we design careful injection of geometric information into disparity features using 3D pose embedding, followed by multi-frame information interaction through an attention module. Subsequently, we encode multi-view relative pose information and geometric relationships between specific sampled points into 3D pose embedding, which is subsequently transferred to the Multi-view Disparity Attention (MDA) module. This method efficiently incorporates the relationship between depth and pixels within the network, facilitating improved information integration across multiple frames. Second, to mitigate the challenge of fluctuating image quality stemming from occlusion and other factors, we maintain and update the disparity hidden features to reflect the depth uncertainty of the current sampling point for each iteration. We design the disparity feature encoding module to learn disparity features along the epipolar lines of multi-view frames. This approach enables us to explicitly characterize occlusion scenarios for each pixel across diverse source images and dynamically adapt them during epipolar disparity flow updates. Consequently, the auxiliary information is furnished for subsequent information fusion within the module. Furthermore, we designed a novel initialization method to further eliminate the influence of the depth range compared to DispMVS [14].

In summary, our contributions can be highlighted as follows: (1) A multi-view disparity transformer network, which facilitates the fusion of information across multi-view frames, (2) A specially designed 3D pose embedding which is utilized to implicitly construct relationships of the epipolar disparity flow among multi-view frames, and (3) An uncertainty estimation module and dynamically updated hidden states representing the quality of source images during iterations. We evaluate our method against other MVS methods on the DTU dataset [23] and Tanks&Temple dataset [24], and demonstrate its generalization in Fig. 1.

## 2 Related Work

### 2.1 Traditional MVS

Multi-View Stereo has been developed for many years and has many downstream or related applications such as simultaneous localization and mapping (SLAM) [25], visual localization [26], 3D reconstruction [27; 28], 3D generation [29] and scene understanding [30]. Traditional methods for Multi-View Stereo (MVS) can generally be categorized into three classes: volumetric, point cloud-based, and 2D depth map-based methods. Volumetric methods [31; 32] typically partition the 3D space into voxels and annotate them as either interior or exterior to the object surface. Point cloud-based methods [33; 34] directly optimize the point cloud coordinates of objects in 3D space. Depth map-based methods [2; 4; 35; 6] first estimate 2D depth corresponding to images and then

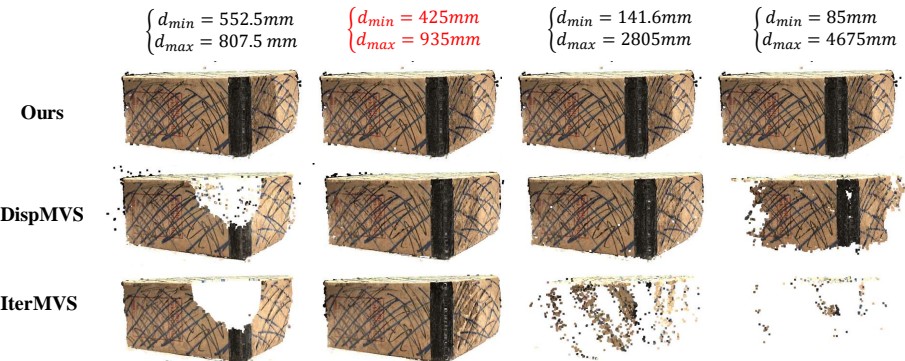

Figure 1: The robustness testing on the depth range. Under identical training configurations, our method exhibits superior robustness to variations in depth range compared with two state-of-the-art methods [13; 14]. The red markings denote the actual depth range used during training.

fuse the 2D depths of the same scene to obtain a 3D point cloud. However, these traditional methods remain constrained by manually crafted image features and similarity matrices.

## 2.2 Deep learning based MVS

**CNN-based MVS methods** generally leverage convolutional neural networks to construct and refine 3D cost volume [36; 37; 11; 38; 39; 40; 41; 42; 43; 44; 45; 46; 47]. For instance, [38] uses isotropic and anisotropic 3D convolution-based learning networks to estimate the depth map. [40] introduces a pixel-wise network to obtain visibility. [11] applies a multi-stage CNN framework to enable reconstruction. [44] and [45] build a kind of pyramid to realize 3D cost volume. Similarly, [46] proposes a sparse-to-dense CNN framework when constructing the 3D cost volume.

**RNN-based methods** mainly exploit recurrent network structures [48; 49] to regularize 3D cost volume [50; 51; 52; 9; 13; 53; 54]. For example, [50] utilizes recurrent encoder-decoder structure and 2D CNN framework to solve large-scale MVS reconstruction. [51] introduces a scalable RNN-based MVS framework. IterMVS [13] uses a GRU-based estimator to encode the probability distribution of depth. Compared with 3D CNN, RNN highly reduces the memory requirement, which makes it more suitable for large-scale MVS reconstruction [55].

**Transformer** is popular in 3D vision tasks [56; 57; 58; 59], and first introduced into the field of MVS reconstruction by [10] due to its ability to capture global context information. Transformer is incorporated into feature encoding [10; 60; 61] to capture features within and between input images. The succeeding work [62] implements a transformer to assign weights to different pixels in the aggregating process. [63] employs an Epipolar Transformer to perform non-local feature augmentation. However, these deep learning-based MVS methods commonly exhibit sensitivity to the depth range, thereby restricting their broad applicability.

**Scale-agnostic MVS methods** infer the depth information from the movement along epipolar lines to reduce the heavy dependence of depth range priors. Several methods [14; 15] perform 2D sampling between two frames and iteratively update flows to find the matching points. Specifically, DispMVS [14] is randomly initialized within the depth range and performs depth fusion by utilizing a weighted sum. RAMDepth [15] selects a random source image in each iteration. However, both methods fail to fully exploit multi-frame constraints during the flow updates due to the mismatch of 3D information at sampling points. In this paper, we enhance the epipolar matching process by simultaneously considering multi-frame information.

## 3 Method

Given a reference image $I_0 \in \mathbb{R}^{H \times W \times 3}$ and multi-view source images $\{I_i\}_{i=1}^{N-1} \in \mathbb{R}^{H \times W \times 3}$ as input, the task of MVS is to calculate the depth map of the reference image. We treat MVS as a matching problem: for a pixel point $p_r$ in the reference image, we identify the corresponding point $p_s$ in the source image, then we can get depth by triangulation. Given the initial matching point $p_s^0$ obtained by the initial depth, we adopt an iterative update strategy. Since the matching point lies on the epipolar line of the source image, the one-degree-of-freedom epipolar disparity flow is used

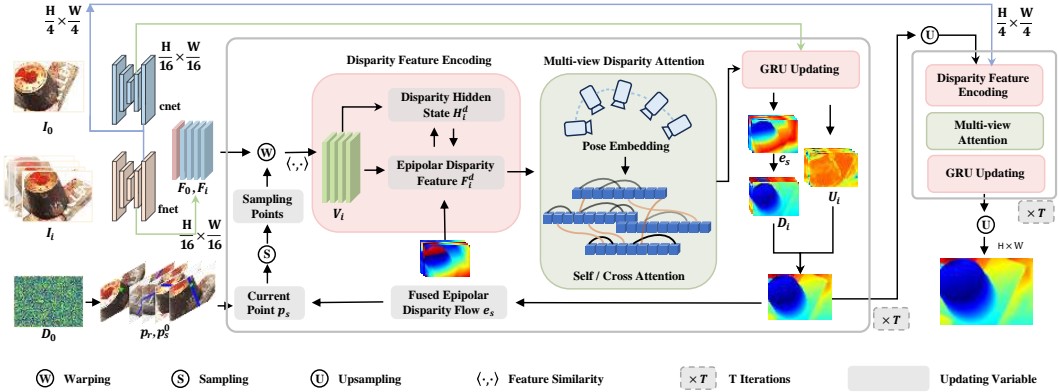

Figure 2: Overview of our method. We introduce the disparity feature encoding module to encode viewpoint quality differences, and the Multi-view Disparity Attention (MDA) module to facilitate information interaction between multi-view images. The MDA module is depicted in Fig. 3. Starting from an initial depth map $D_0$, the epipolar disparity flows are iteratively updated and fused to the depth of the next stage.

to represent the network's iterative updates. The epipolar disparity flow $e_s^k \in \mathbb{R}^{H \times W \times 1}$ is 1-d flow along the epipolar line on the source image during each iteration:

$$e_s^k = \vec{e_{dir}} \cdot (p_s^k - p_s^0), \tag{1}$$

where $\vec{e_{dir}}$ is the normalized direction vector of the epipolar line, $\cdot$ is the dot product of vectors, and $k$ is the iteration time. Different from previous methods [14; 13], we fully eliminate the dependence on depth range during initialization and achieve synchronous updating of the epipolar disparity flow across multi-view images. This is done by our design of disparity information interaction.

The overall pipeline of our method is illustrated in Fig. 2. The proposed method starts from a feature extraction module to extract multi-scale features (Sec. 3.1). Then, we discuss how to initialize the depth map without depth range (Sec. 3.2) and perform feature encoding (Sec. 3.3). To facilitate information fusion across multi-view source images, we introduce the Multi-view Disparity Attention (MDA) module (Sec. 3.4), enhanced with Pose Embedding. Finally, the features enhanced by the MDA module are fed into a GRU module to update the epipolar disparity flow, as described (Sec. 3.5), which is then fused to generate the depth map.

## 3.1 Feature Extraction

Following previous methods [10; 13; 60; 64], we employ convolutional neural networks (CNNs) for image feature extraction. Moreover, we adopt a coarse-to-fine strategy to extract multi-scale image features. Specifically, we utilize two share-weighted feature extraction modules to extract image features $F_0^l \in \mathbb{R}^{H \times W \times C}$ and $\{F_i^l\}_{i=1}^{N-1} \in \mathbb{R}^{H \times W \times C}$ and a context feature extraction module to extract context features.

## 3.2 Initialization

Differing from DispMVS [14], we design a novel initialization method without depth range to further mitigate the influence of depth priors. Specifically, we select an initial position along the epipolar line and then convert it into the depth map. First, we derive the correspondence between depth and position along the epipolar line. Given a pixel $p_r$ of the reference image $I_0$, the geometric constrain between it and the warped pixel $p_s$ of the source image $I_i$ can be written as:

$$K_i^{-1} p_s d_s = R \cdot \left( K_0^{-1} p_r d_r \right) + T, \tag{2}$$

where $d_r$ denotes the depth in reference view, $d_s$ denotes the depth in source view. $R$ and $t$ denote the rotation and translation between the reference and the source view, $K_0^{-1}$ and $K_i^{-1}$ denote the intrinsic matrices of the reference and the source view. Let $T = (t_x, t_y, t_z)^T$, $K_i^{-1} p_s = (p_{sx}, p_{sy}, p_{sz})^T$ and $R \cdot K_0^{-1} p_r = (p_{rx}, p_{ry}, p_{rz})^T$, we can associate $d_r$ and $d_s$ with pixel coordinates:

$$d_r = \begin{cases} (t_x p_{sz} - t_z p_{sx})/(p_{sx} p_{rz} - p_{sz} p_{rx}), & \left| \overrightarrow{f}_{xr \to xs}(p_r) \geq \overrightarrow{f}_{yr \to ys}(p_r) \right| \\ (t_y p_{sz} - t_z p_{sy})/(p_{sy} p_{rz} - p_{sz} p_{ry}), & \text{otherwise} \end{cases} \tag{3}$$

$$d_s = \begin{cases} (t_x p_{rz} - t_z p_{rx})/(p_{sx} p_{rz} - p_{sz} p_{rx}), & \left| \overrightarrow{f}_{xr \to xs}(p_r) \geq \overrightarrow{f}_{yr \to ys}(p_r) \right| \\ (t_y p_{rz} - t_z p_{ry})/(p_{sy} p_{rz} - p_{sz} p_{ry}), & \text{otherwise} \end{cases} \tag{4}$$

where $\overrightarrow{f}$ is a 2D flow vector along the epipolar line that provides flow in the x dimension $\overrightarrow{f}_{xr \to xs}(p_r)$ and y dimension $\overrightarrow{f}_{yr \to ys}(p_r)$. To obtain an appropriate initial position, we first determine the geometrically valid range along the epipolar line, which has not been considered in other works [53; 14]. If a point is observable in the current view, it must have physical significance, meaning it must lie in front of the camera. Therefore, we identify the search range along the epipolar line on the source image that satisfies the condition $d_r > 0, d_s > 0$. We obtain the initial position $p_s^0$ by selecting the mid-point in search range along epipolar line.

### 3.3 Disparity Hidden State Based Feature Encoding

Due to occlusion, moving objects, blurring, or other factors violating the multi-view geometry assumptions, the quality of sampling points from different source images varies, which limits the network's performance in depth estimation. To address this issue, we extract uncertainty information from the sampling point feature and encode it with cost volume as epipolar disparity feature $F_i^d$. As shown in Fig. 2, we design the disparity hidden state $H_i^d$ to maintain the sampling information of the current source image and update it during iterations by incorporating new uncertainty information.

**Cost Volume Construction.** For each source image, after determining the position $p_s^t$ for the current iteration, we uniformly sample $M$ points around $p_s^t$ along the epipolar line at each scale with a distance of one pixel. By constructing a 4-layer pyramid feature using average pooling, uniform pixel sampling at different levels allows for a larger receptive field. The sampling interval in 2D is fixed. Given image features $F_0^l$ and $\{F_i^l\}_{i=1}^{N-1}$, we obtain the features of $M$ sampled points in the source image through interpolation and calculate the visual similarity. The cost volume $V \in \mathbb{R}^{H \times W \times M}$ is constructed by computing the dot product between pairs of image feature vectors:

$$V_i(p_r) = \sum_{r \in R} \left\langle F_{I_0}(p_r) \cdot F_{I_i}\left(p_s^k + r\right) \right\rangle, \tag{5}$$

where $R$ represents the set of sampling points uniformly sampled along the epipolar line in the source image, and $M$ denotes the number of sample points.

**Disparity Feature Encoding with Uncertainty.** When estimating the epipolar disparity flow from multi-view frames, it is essential to encode the differences between source images caused by variations in occlusion situations and image quality. Motivated by this, we conduct disparity hidden state $H_i^d \in \mathbb{R}^{H \times W \times C}$ to explicitly represent the situation of point $p_r$ relative to the source image. Motivated by this, we introduce a disparity hidden state $H_i^d \in \mathbb{R}^{H \times W \times C}$ to explicitly represent the condition of points relative to the multi-view source images. $H_i^d$ is randomly initialized and consecutively updated throughout the iterative process. We introduce a variance-based uncertainty estimation module to encode the correlation features, which is formulated as follows:

$$U_i = 1 - \sigma \left( \sum \left(V_i - \overline{V_i}\right)^2 \right), \tag{6}$$

where $V_i$ denotes the cost volume of source image, $\overline{V_i}$ denotes the average value of $V_i$, and $\sigma(\cdot)$ is the sigmoid function. Then, the uncertainty $U_i$, the disparity hidden state of the previous iteration, the correlation features and the epipolar disparity flows are fed into the convolutional layers to generate epipolar disparity feature $F_i^d$ and update the disparity hidden state $H_i^d$.

### 3.4 Multi-view Stereo Transformer

DispMVS estimates the epipolar disparity flow from each two-frame image pair $\{I_0, I_i\}_{i=1}^{N-1}$, which overlooks the abundant multi-view information. Inspired by VideoFlow [21], we estimate the epipolar

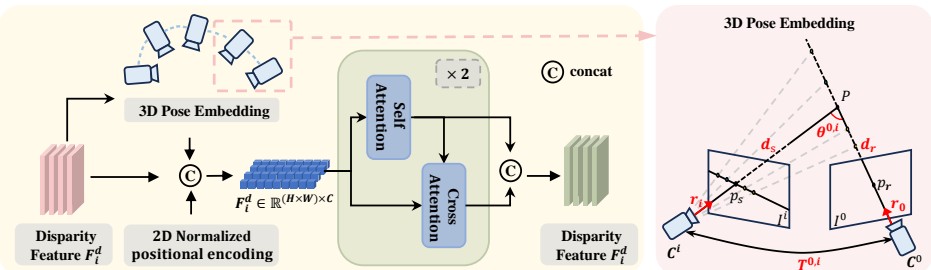

Figure 3: Illustration of MDA module. After concatenating features with 3D pose embedding and 2D normalized positional encoding, we achieve intra-image and inter-image information interaction through self-attention and cross-attention. As shown in the right figure, 3D pose embedding encodes relative pose and pixel geometric information into the features to enhance the learning capability of the attention mechanism.

flow of multi-view images simultaneously. However, since multiple source images are not sequentially arranged and points uniformly 2D sampled across source images can not establish robust 3D spatial correspondences, directly learning the continuity between flows, as [21], does not work.

Therefore, unlike [13; 10], etc., we design some special structures for information aggregation among multi-view images. Although the depths of sampled points along epipolar lines do not correspond, we observe that there is a regular pattern in the direction of depths along epipolar lines. As shown in Fig. 3, we design Multi-view Disparity Attention to learn the global information and utilize pose embedding to implicitly model the correspondence between pixel coordinates and depth on multiple source images, enabling the network to learn the direction and scale relationship of corresponding flows across different source images.

**Multi-view Disparity Attention.** To effectively capture extensive global information across epipolar disparity features from different views, we leverage the Multi-view Disparity Attention (MDA) module to further enhance the disparity features. We utilize an attention module to globally interact with disparity features of multi-view source images, thereby achieving multi-view feature fusion.

Given epipolar disparity features $\{F_i^d\}_{i=1}^{N-1}$, we first use self-attention to achieve intra-image information interaction. We concatenate epipolar disparity features $F_i^d \in \mathbb{R}^{H \times W \times C}$ and set $H \times W$ the as sequence length $L$, generating $F^d \in \mathbb{R}^{(N-1) \times (H \times W) \times C}$.

Then we use cross-attention to achieve inter-frame information interaction and learn the relations among multi-view. We concatenate epipolar disparity features $F_i^d$ and set the number of source images $N-1$ the as sequence length $L$, generating $F^d \in \mathbb{R}^{(H \times W) \times (N-1) \times C}$.

To reduce computation cost, for the self-attention we use a linear transformer to compute attention, which replaces the original kernel function with:

$$\text{Attention}(Q, K, V) = \Phi(Q)\left(\Phi\left(K^T\right)V\right), \tag{7}$$

where $\Phi(\cdot) = \text{relu}(\cdot) + 1$ and $\text{relu}(\cdot)$ represents the activation function of exponential linear units.

**Pose Embedding.** Due to the depths of sampling points varying for different source images, we utilize pose embedding to construct implicit disparity relationships among multi-view frames. To effectively convey useful information to the attention module, we categorize the features of pose embedding into two types: multi-view relative pose information and geometric information between specific sampled points. Fig. 3 illustrates the variables used to construct the pose embedding.

On one hand, the multi-view relative pose information between cameras contains crucial information about disparity features. By explicitly injecting relative poses into the attention module, the network can learn image-level geometric constraints. We represent the angle $\theta^{0,i}$ between rays as embedding. Inspired by [65], we encode the rotation matrix and translation matrix between the reference and the source view into the relative pose distance $P^{0,i}$:

$$P^{0,i} = \sqrt{\|t^{0,i}\| + \frac{2}{3}tr\left(\mathbb{I} - R^{0,i}\right)}, \tag{8}$$

On the other hand, we encode the geometric information between specific sampled points. Due to our incorporation of pixel-level attention in addition to inter-frame attention, it is necessary to encode not

only image-level camera poses but also the pixel-level information corresponding to sampled points. It is important to note that for each pixel in the reference image and its corresponding sampled point in the source image, we can obtain the corresponding 3D point coordinates $P$ through triangulation based on stereo geometry. Accordingly, we encode the 2D coordinates $p_s$ of the source image, the depth $d_r$ from the perspective of the reference image, and the depth $d_s$ from the perspective of the source image, thereby transforming the 3D information into corresponding relationships on the 2D plane. Moreover, we encode the normalized direction $r_0, r_i$ to the 3D location of a point.

### 3.5 Iterative Updates

In the GRU updating process, we iteratively update the epipolar disparity flow $e_s^{k+1} = e_s^k + \Delta e_s$ obtained from the MDA module for each source image. In each iteration, the input to the update operator includes 1) the hidden state; 2) the disparity feature output from the MDA module; 3) the current epipolar flow; and 4) the context feature of the reference image. The output of the update operator includes 1) a new hidden state; 2) an increment to the disparity flow; and 3) the weight of disparity flow for multi-view images. We derive the depth from the disparity flow and employ a weighted sum to integrate the depth across multi-view source images. After fusion, the depth is converted back to disparity flow to perform the next iteration.

### 3.6 Loss Function

Similar to [14], we output depth after each iteration and construct the loss function accordingly. We construct the depth L1 loss. The loss function is represented in Eq. 9:

$$loss = \sum_{j=t_c, t_f} \sum_{0<=k<j} \gamma^k |\mathrm{norm}\left(gt_r\right) - \mathrm{norm}\left(d_r^k\right)|, \tag{9}$$

where $t_c$, $t_f$ are iterations at the coarse and fine stage, $\gamma$ is a hyper-parameter which is set to 0.9.

## 4 Experiments

In this section, we first introduce the datasets (Sec. 4.1), followed by the implementation details of the experiment (Sec. 4.2). Subsequently, we delineate the experimental performance (Sec. 4.3) and conduct ablation experiments to validate the efficacy of each proposed module (Sec. 4.4).

### 4.1 Datasets

DTU dataset [23] is an indoor multi-view stereo dataset captured in well-controlled laboratory conditions, which contains 128 different scenes with 49 views under 7 different lighting conditions. Following MVSNet [8], we partitioned the DTU dataset into 79 training sets, 18 validation sets, and 22 evaluation sets. BlendedMVS dataset [66] is a large-scale outdoor multi-view stereo dataset that contains a diverse array of objects and scenes, with 106 training scenes and 7 validation scenes. Tanks and Temples [24] is a public multi-view stereo benchmark captured under outdoor real-world conditions. It contains an intermediate subset of 8 scenes and an advanced subset of 6 scenes.

### 4.2 Implementation Details

Implemented by PyTorch [67], two models are trained on the DTU dataset and large-scale Blended-MVS dataset, respectively. On the DTU dataset, we set the image resolution as $640 \times 512$ and the number of input images as 5 for the training phase. On the BlendedMVS dataset, we set the image resolution as $768 \times 576$ and the number of input images as 5 for the training phase. For all models, we use the AdamW optimizer with an initial learning rate of 0.0002 that halves every four epochs for 16 epochs. The training procedure is finished on two A100 with $t_c = 8$, $t_f = 2$. For depth filtering and fusion, we process 2D depth maps to generate point clouds and compare them with ground truth.

### 4.3 Experimental Performance

In this section, we compare our method with other state-of-the-art methods and scale-agnostic methods. Existing methods are categorized into traditional methods [2; 35], 3D cost-volume

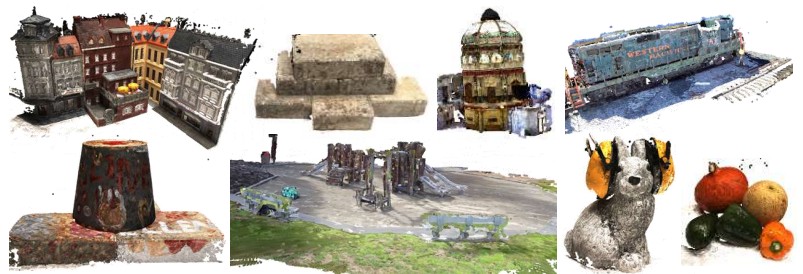

Figure 4: Some qualitative results of the proposed method on DTU and Tanks and Temples datasets.

Table 1: The Quantitative point cloud evaluation results on DTU evaluation set. The lower the Accuracy (Acc), Completeness (Comp), Overall, the better. We split methods into four categories and highlight the best in bold for each.

| Method | ACC.(mm)↓ | Comp.(mm)↓ | Overall(mm)↓ |
|---|---|---|---|
| Gipuma [2] | **0.283** | 0.873 | 0.578 |
| COLMAP [35] | 0.400 | **0.664** | **0.532** |
| MVSNet [8] | 0.396 | 0.527 | 0.462 |
| AA-RMVSNet [9] | 0.376 | 0.339 | 0.357 |
| PatchmatchNet [53] | 0.427 | 0.277 | 0.352 |
| UniMVSNet [41] | 0.352 | 0.278 | 0.315 |
| TransMVSNet [10] | 0.321 | 0.289 | 0.305 |
| MVSTER* [68] | 0.340 | 0.266 | 0.303 |
| GeoMVS [69] | 0.331 | 0.259 | 0.295 |
| GBiNet [47] | 0.315 | 0.262 | 0.289 |
| MVSFormer++[71] | **0.309** | **0.252** | **0.281** |
| IterMVS [13] | 0.373 | 0.354 | 0.363 |
| CER-MVS [70] | **0.359** | **0.305** | **0.332** |
| RAMDepth [15] | 0.447 | **0.278** | 0.362 |
| DispMVS [14] | 0.354 | 0.324 | 0.339 |
| Ours | **0.338** | 0.331 | **0.335** |

methods[8; 53; 9; 68; 47; 10; 69; 61], RNN-based methods [13; 70] and scale-agnostic methods [15; 14]. Methods that leverage scene depth range have an advantage as they can utilize accurate and robust information, thereby mitigating outliers, especially in textureless regions.

**Evaluation on DTU.** We evaluate the proposed method on the evaluation set of DTU dataset. We set the image resolution as $1600 \times 1152$ and the number of input images as 5. As shown in Table 1, our method has the best overall performance among depth-range-free methods. CER-MVS [70] and MVSFormer++ [61] demonstrate superior performance; however, they are heavily dependent on the accuracy of the depth range. Our approach outperforms when compared with depth range-free methods like DispMVS [14] and RAMDepth [15], which demonstrates the effectiveness of our method in exploiting correlations among multi-view frames.

**Evaluation on Tanks and Temples.** Since the Tanks and Temples dataset does not provide training samples, we use a model pre-trained on the BlendedMVS dataset for testing. We set the image resolution as $1920 \times 1024$ and the number of input images as 7 for the evaluation phase. Table 2 presents the comparison between our method and other state-of-the-art methods. Our method achieves the best performance among scale-agnostic methods [14]. Since RAMDepth [15] has not provided results on the Tanks and Temples dataset and source code, we are unable to make a comparison. Although our method exhibits a certain gap when compared to state-of-the-art methods [70; 61] based on precise depth priors, it demonstrates superior robustness across a broader depth range.

We visualize point clouds generated on DTU and Tanks and Temples dataset in Fig. 4, which demonstrates that our method is capable of constructing a comprehensive and precise point cloud.

## 4.4 Ablation Study

In this subsection, we conduct ablation studies of our model trained on DTU [23] datasets to discuss the effectiveness of core parts of our method. The implemented baseline is basically based on DispMVS [14]. All the experiments are performed with the same hyperparameters.

Table 2: The Quantitative point cloud evaluation results on Tanks and Temples benchmark. The metric is F-score and "Mean" refers to the average F-score of all scenes (higher is better). We split methods into four categories and highlight the best in bold for each.

| Method | advanced | | | | | | | intermediate | | | | | | | | |
|---|---|---|---|---|---|---|---|---|---|---|---|---|---|---|---|---|
| | Mean | Aud. | Bal. | Cou. | Mus. | Pal. | Tem. | Mean | Fam. | Fra. | Hor. | Lig. | M60 | Pan. | Pla. | Tra. |
| COLMAP [35] | 27.24 | 16.02 | 25.23 | 34.70 | 41.51 | 18.05 | 27.94 | 42.14 | 50.41 | 22.25 | 26.63 | 56.43 | 44.83 | 46.97 | 48.53 | 42.04 |
| MVSNet [8] | - | - | - | - | - | - | - | 43.48 | 55.99 | 28.55 | 25.07 | 50.79 | 53.96 | 50.86 | 47.90 | 34.69 |
| PatchmatchNet [53] | 32.31 | 23.69 | 37.73 | 30.04 | 41.80 | 28.31 | 32.29 | 53.15 | 66.99 | 52.64 | 43.24 | 54.87 | 52.87 | 49.54 | 54.21 | 50.81 |
| AA-RMVSNet [9] | 33.53 | 20.96 | 40.15 | 32.05 | 46.01 | 29.28 | 32.71 | 61.51 | 77.77 | 59.53 | 51.53 | 64.02 | 64.05 | 59.47 | 60.85 | 54.90 |
| MVSTER [68] | 37.53 | 26.68 | 42.14 | 35.65 | 49.37 | 32.16 | 39.19 | 60.92 | 80.21 | 63.51 | 52.30 | 61.38 | 61.47 | 58.16 | 58.98 | 51.38 |
| GBi-Net [47] | 37.32 | 29.77 | 42.12 | 36.30 | 47.69 | 31.11 | 36.93 | 61.42 | 79.77 | 67.69 | 51.81 | 61.25 | 60.37 | 55.87 | 60.67 | 53.89 |
| TransMVSNet [10] | 37.00 | 24.84 | 44.59 | 34.77 | 46.49 | 34.69 | 36.62 | 63.52 | 80.92 | 65.83 | 56.94 | 62.54 | 63.06 | 60.00 | 60.20 | 58.67 |
| GeoMVSNet [69] | 41.52 | 30.23 | **46.53** | **39.98** | 53.05 | **35.98** | 43.34 | 65.89 | 81.64 | 67.53 | 55.78 | 68.02 | **65.49** | **67.19** | **63.27** | 58.22 |
| MVSFormer++ [61] | **41.70** | **30.39** | 45.85 | 39.35 | **53.62** | 35.34 | **45.66** | 67.03 | 82.87 | 68.90 | 64.21 | 68.65 | 65.00 | 66.43 | 60.07 | **60.12** |
| IterMVS [13] | 33.24 | 22.95 | 38.74 | 30.64 | 43.44 | 28.39 | 35.27 | 56.94 | 76.12 | 55.80 | **50.53** | 56.05 | 57.68 | 52.62 | 55.70 | 50.99 |
| CER-MVS [70] | **40.19** | **25.95** | **45.75** | **39.65** | **51.75** | **35.08** | **42.97** | 64.82 | 81.16 | 64.21 | 50.43 | **70.73** | 63.85 | 63.99 | 65.90 | 58.25 |
| DispMVS [14] | 34.90 | 26.09 | 38.01 | 33.19 | 44.90 | 28.49 | 38.75 | 59.07 | 74.73 | 60.67 | **54.13** | 59.58 | 58.02 | **53.39** | 58.63 | 53.42 |
| Ours | **35.95** | **26.75** | **40.22** | **33.87** | **45.78** | **29.58** | **39.50** | 59.27 | **75.05** | **61.63** | 53.15 | **60.24** | **58.44** | 53.34 | **58.79** | **53.54** |

Table 3: Quantitative evaluation of the effectiveness of each component of the model was conducted on the DTU dataset. The lower the Accuracy (Acc), Completeness (Comp), Overall, the better.

| Pose Embedding | Uncertainty | Disparity Hidden State | Acc.(mm)↓ | Comp.(mm)↓ | Overall(mm)↓ |
|---|---|---|---|---|---|
| ✓ | | | 0.363 | 0.382 | 0.372 |
| | ✓ | ✓ | 0.356 | 0.384 | 0.370 |
| ✓ | ✓ | | 0.370 | 0.354 | 0.362 |
| ✓ | | ✓ | **0.336** | 0.378 | 0.357 |
| ✓ | ✓ | ✓ | 0.338 | **0.331** | **0.335** |

**Pose Embedding.** We conducted ablation experiments to validate the effectiveness of the pose embedding. Specifically, within the multi-view attention module, we remove 3D pose embedding and retain only the original 2D position encoding of attention. As shown in Table 3, after applying pose embedding, the overall performance improves by 9.46%. The result indicates that the current task heavily relies on the relative pose and geometric information contained in the pose embedding. Without incorporating geometric constraints across multi-view source images, typically achieved through pose embedding, the performance of Transformer in this task may degrade significantly.

**Disparity Feature Encoding coupling with Uncertainty.** Following [14], we further attempt to remove uncertainty estimation and disparity hidden state and directly perform feature encoding on disparity flow and cost volume. As shown in Table 3, with the disparity feature encoding coupling with uncertainty, the overall performance improves by 9.95%. The result validates the effectiveness of the module, demonstrating that explicitly estimating the quality of sampled points on the epipolar line of source images and updating the disparity hidden state in the network is effective. Additionally, we designed two separate ablation experiments, removing the uncertainty and disparity feature hidden states, to further evaluate the impact of these two modules on the network. The uncertainty and disparity feature hidden states improved the overall performance by 6.16% and 7.46%, respectively. Compared to the performance without disparity feature encoding coupling with uncertainty, this demonstrates the effectiveness of the uncertainty and disparity feature hidden state updating modules.

## 4.5 Depth Range

In this section, we compare the generalization of different networks to depth range. We don't compare with RAMDepth [15] in the ablation studies due to the lack of its source code. We select several state-of-the-art methods (GeoMVS [69], MVSFormer++ [61]), RNN-based methods (IterMVS [13], CER-MVS [70]) and depth-range-free method (DispMVS [14]) to conduct experiments to evaluate the generalization of depth range. Our main comparison is with depth-range-free methods [14], which reduce dependence on depth priors through network design. All methods are trained on DTU dataset with the same depth range and subsequently inference under different depth ranges.

Table 4: Quantitative evaluation of the sensitivity of methods to depth range. The results for Accuracy (Acc), Completeness (Comp), and Overall are presented in millimeters (mm).

| Depth Range | Method | Acc.↓ | Comp.↓ | Overall↓ |
|---|---|---|---|---|
| (425mm, 935mm) | IterMVS[13] | 0.373 | 0.354 | 0.363 |
| | DispMVS[14] | 0.354 | 0.324 | 0.339 |
| | CER-MVS[70] | 0.359 | 0.305 | 0.332 |
| | GeoMVS[69] | 0.331 | 0.259 | 0.295 |
| | MVSFormer++[71] | **0.309** | **0.252** | **0.281** |
| | Ours | 0.338 | 0.331 | 0.335 |
| (212.5mm, 1870mm) | IterMVS[13] | 0.532 | 1.471 | 1.002 |
| | DispMVS[14] | 0.348 | 0.404 | 0.376 |
| | CER-MVS[70] | 1.805 | 1.161 | 1.483 |
| | GeoMVS[69] | 0.435 | 0.619 | 0.527 |
| | MVSFormer++[71] | 0.478 | 0.359 | 0.418 |
| | Ours | **0.338** | **0.331** | **0.335** |
| (141.6mm, 2805mm) | IterMVS[13] | 0.935 | 6.985 | 3.960 |
| | DispMVS[14] | **0.314** | 0.671 | 0.493 |
| | CER-MVS[70] | 11.464 | 12.683 | 12.073 |
| | GeoMVS[69] | 0.602 | 1.663 | 1.133 |
| | MVSFormer++[71] | 0.739 | 0.820 | 0.780 |
| | Ours | 0.338 | **0.331** | **0.335** |

Table 5: The influence of the depth range obtained from COLMAP of methods. The results for Accuracy (Acc), Completeness (Comp), and Overall are presented in millimeters (mm).

| Depth Range | Method | Acc.↓ | Comp.↓ | Overall↓ |
|---|---|---|---|---|
| GT | IterMVS[13] | 0.373 | 0.354 | 0.363 |
| | DispMVS[14] | 0.354 | 0.324 | 0.339 |
| | CER-MVS[70] | 0.359 | 0.305 | 0.332 |
| | GeoMVS[69] | 0.331 | 0.259 | 0.295 |
| | MVSFormer++[71] | **0.309** | **0.252** | **0.281** |
| | Ours | 0.338 | 0.331 | 0.335 |
| COLMAP | IterMVS[13] | 0.454 | 1.486 | 0.970 |
| | DispMVS[14] | 0.339 | 0.372 | 0.356 |
| | CER-MVS[70] | 0.816 | 0.326 | 0.571 |
| | GeoMVS[69] | 0.374 | 0.415 | 0.394 |
| | MVSFormer++[71] | 0.361 | **0.319** | 0.340 |
| | Ours | **0.338** | 0.331 | **0.335** |

For methods [13; 70; 69; 61] that rely on depth range prior for depth sampling, whether based on RNN or Transformer, they may exhibit better performance with accurate depth priors. However, as shown in Table 4, there is a marked decline in performance for these methods with larger depth range. Although DispMVS [14] showed insensitivity to depth range, its performance still exhibited a certain degree of decline with larger depth ranges. In contrast, our method, which is independent of depth range, maintained consistent performance regardless of changes in depth range.

It is crucial to emphasize that the depth range provided by the dataset is exceptionally accurate. For instance, the ground truth for the Tanks-and-Temples dataset is captured using an industrial laser scanner. However, in practical applications, while Structure-from-Motion (SfM) can derive depth ranges from sparse feature points, the resulting depth estimates are often prone to inaccuracies. These inaccuracies arise from the inherent sparsity of feature points, as well as challenges such as occlusion and suboptimal viewpoint selection. To verify the robustness of the MVS models in practical applications, we use the depth range obtained from COLMAP to replace the depth range ground truth (GT). As shown in Table 5, there is a significant decline in performance for GeoMVS [69], MVSFormer++ [61], IterMVS [13] and CER-MVS [70] when we use the depth range obtained from COLMAP. DispMVS [14] also exhibits a certain degree of decline. In contrast, our method maintained consistent performance. This result further demonstrates the necessity of eliminating the depth range.

## 5    Conclusion

We propose a prior-free multi-view stereo framework that simultaneously considers all the source images. To fully fuse the information from disordered and arbitrarily posed source images, we propose a 3D-pose-embedding-aided and uncertainty-driven transformer-based network. Extensive experiments show that our methods achieve state-of-the-art performances among the prior-free methods and exhibit greater robustness to the depth range prior. **Limitations**: The proposed method cannot run in real-time (i.e., 30 FPS), which could limit its application in mobile devices or other time-sensitive scenarios. Besides, our method shows a performance gap compared to SOTA cost-volume-based methods on the mainstream benchmark, despite these methods relying on highly precise depth range priors. In the future work, we hope to close the gap.

## Acknowledgement

This work was partially supported by NSF of China (No. 61932003).

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
