# OpenReview forum: "A Global Depth-Range-Free Multi-View Stereo Transformer Network with Pose Embedding"
_NeurIPS.cc/2024/Conference — NeurIPS 2024 poster_

### Official Review · Reviewer_s1Ui · 2024-07-07

**Soundness:** 3
**Presentation:** 2
**Contribution:** 2
**Rating:** 6
**Confidence:** 4

**Summary:**

This paper proposes a depth-range free MVS network that considers the information within the MVS framework. The initial depth is derived from the epipolar geometry of the reference image and the source images. In addition, the disparity features are enhanced by an uncertainty encoding module and a multi-view disparity attention module. The proposed method is assessed on several benchmarks compared to existing works. It shows promising results regarding the robustness to different depth ranges and the overall performance compared to RNN-based methods.

**Strengths:**

1. The evaluation of the proposed method shows better results than the existing RNN-based methods on several public benchmarks and it is more robust to different depth ranges.

**Weaknesses:**

1. The methodology lacks some clarity. For example, a) no explanation for $\vec{f}$ in equations 3 and 4, b) no explanation for $0$ and $1$ in $p^{0}$ and $p^{1}$ for the search range, c) the dot before $\varphi4$ in equation 8, d) how $H_{di}$ is initialized in equation 8, e) in equation 10, is $R^{0,i}$ the relative rotation between two images, or generated by the intersection angle $\theta^{0,1}$, f) no explanation for $t_{c}$ and $t_{f}$ in equation 11. Readers in the same research field might get the actual meaning without indication, but the broader audiences need to be considered.

2. There are problems with writing. For example, a) wrong reference for DispMVS, b) table 2 is indexed before tables 3 and 4 but it is mentioned after them in the main text, 3) missing reference of a section in the last paragraph in section 1.

**Questions:**

1. In Figure 1, when the depth range is clipped within the actual depth range, e.g., the first column, how can you still get the depth for the foreground (425mm -- 552.5mm) and background (807.5mm -- 935mm) points? It seems that the proposed method yields the same results for all configurations as the four figures look the same.

2. When sampling M points to construct the cost volume, what is the interval between the points and how it is decided? Is it adaptive during the iterations?

**Limitations:**

1. It is suggested that the authors refine the writing of the methodology section and make clarifications regarding W1 and Q2.

---

> ### Author Rebuttal · Authors · 2024-08-06
>
> > no explanation for $\overrightarrow{f}$ in equations 3 and 4.
>
> Similar to DispMVS, $\overrightarrow{f}$ is a 2D flow vector along the epipolar line that provides flow in the x dimension $\overrightarrow{f} _{xr\to xs }(p_r)$ and y dimension $\overrightarrow{f} _{yr\to ys }(p_r)$.
>
> > no explanation for 0 and 1 in $p_0$ and $p_1$ for the search range.
>
> In Sec 3.2 we define the search range as $[p_{d_0,d_i>0}^0, p_{d_0,d_i>0}^1 ]$, where 0 and 1 represent the left end and right end of the range.
>
> > the dot before $\varphi_4$ in equation 8.
>
> We will remove the redundant dot before $\varphi_4$ in Eq. 8.
>
> > how $Hd_i$ is initialized in equation 8.
>
> $Hd_i$ is randomly initialized and learned via training.
>
> > in equation 10, is $R^{0,i}$ the relative rotation between two images, or generated by the intersection angle $\theta ^{0,i}$.
>
> As mentioned in paper, the calculation of relative pose distance is exactly the same as in [12], where $R^{0,i}$ is the relative rotation between two images.
>
> > no explanation for $t_c$ and $t_f$ in equation 11.
>
> Similar to DispMVS, $t_c$ represents the iterations at the coarse stage, and $t_f$ represents the iterations at the fine stage.
>
> > wrong reference for DispMVS
>
> We will correct the citation accordingly.
>
> > table 2 is indexed before tables 3 and 4 but it is mentioned after them in the main text.
>
> This issue is due to the LaTeX formatting, and we will make the necessary modifications.
>
> > missing reference of a section in the last paragraph in section 1.
>
> Thank you for your suggestion. We will make the necessary modifications.
>
> > In Figure 1, when the depth range is clipped within the actual depth range, how can you still get the depth for the foreground (425mm -- 552.5mm) and background (807.5mm -- 935mm) points?
>
> Since we have completely removed the depth range prior, the output results are not affected by any errors in the depth range.
> In contrast, IterMVS samples based on the depth range, which prevents it from obtaining the correct depth for foreground (425mm - 552.5mm) and background (807.5mm - 935mm) points.
> DispMVS's initialization also rely on the depth range. During each iteration, it uses the depth range to apply a depth
> normalization and filter out outliers for stability, so an underestimated depth range can significantly affect its performance.
>
> > What is the interval between the sampling points and how it is decided? Is it adaptive during the iterations?
>
> Similar to DispMVS, we perform 2D sampling, where we sample
> $M$ points around current position $p_i$ along the epipolar line at each scale with a distance of one pixel. As noted on L.169, by constructing a 4-layer pyramid feature using average pooling, uniform pixel sampling at different levels allows for a larger receptive field. The sampling interval in 2D is fixed.

---

> > ### Comment · Reviewer_s1Ui · 2024-08-10
> >
> > Thanks for the responses and clarification.

---

### Official Review · Reviewer_Q3xc · 2024-07-12

**Soundness:** 2
**Presentation:** 1
**Contribution:** 3
**Rating:** 5
**Confidence:** 5

**Summary:**

This paper proposes a depth-range-free Multi-View Stereo (MVS) method, which iteratively updates the depth using a GRU-based approach. To eliminate the dependency on depth priors, the paper improves the depth initialization method of DispMVS. To fully utilize multi-frame information, the paper encodes the observation information between multiple source frames and the reference frame into features and proposes a Multi-view Disparity Attention module for multi-frame information fusion. To enhance the preservation and utilization of geometric information, the paper introduces 3D pose embedding, uncertainty estimation, and disparity hidden states. The paper has been tested on the DTU and T&T datasets, and the experimental results show that the method exhibits robust depth estimation results in the absence of depth range priors.

**Strengths:**

-The motivation for the model design is clear, and the experimental results to some extent reflect the effectiveness of the model.

-The experimental results demonstrate that the method can achieve good predictive results without relying on depth priors.

**Weaknesses:**

-Poor writing quality: The consistency of the symbols in the paper is poor, such as the left side of Eq. 5 should be V_i(p0); The Fd_i in Fig. 2, Eq. 8, and F^d_i in Sec 3.4, the H_i in Fig. 2 and  Hd_i in Eq. 8; the use of symbols is not standardized, such as the representation of matrices and vectors, the authors are advised to carefully check the entire text; In Eq. 8, there is an extra dot multiplication symbol before phi_4.

-Missing method description: For the final GRU update part, the paper lacks a complete description. My current guess is that the authors use Fd_i and context features as input information to update the hidden state not mentioned in the paper, and decode the residual from this hidden state.

-Incorrect key citation: There is an error in the citation of one of the main comparative methods, DispMVS[5], and the authors are advised to carefully check the reference list. "Rethinking the multi-view stereo from the perspective of rendering-based augmentation." -> "Rethinking Disparity: A Depth Range Free Multi-View Stereo Based on Disparity"

-The application of cross-attention is strange: The cross-attention in this paper is only calculated at the same pixel positions in different frames, and the paper emphasizes that there are few continuous smooth trajectories between different frames in MVS, making this design very illogical. The authors are expected to provide a detailed explanation.

-Missing comparisons: CER-MVS [Ⅰ], as an early GRU-based MVS work, should be included in the comparison. Other works such as MVSFormer series[Ⅱ, Ⅲ], GeoMVSNet [Ⅳ] should not be ignored either.

-Incomplete ablation experiments: There is a lack of separate ablation experiments on uncertainty and disparity feature hidden states, and the existing ablation experiments couple the two together. There is a lack of ablation experiments on the MDA module.

[Ⅰ] Ma, Zeyu, Zachary Teed, and Jia Deng. "Multiview stereo with cascaded epipolar raft." ECCV 2022.

[Ⅱ] Cao, Chenjie, Xinlin Ren, and Yanwei Fu. "MVSFormer: Multi-view stereo by learning robust image features and temperature-based depth." arXiv preprint arXiv:2208.02541 (2022).

[Ⅲ] Cao, Chenjie, Xinlin Ren, and Yanwei Fu. "MVSFormer++: Revealing the Devil in Transformer's Details for Multi-View Stereo." ICLR 2024.

[Ⅳ] Zhang, et al. "Geomvsnet: Learning multi-view stereo with geometry perception." CVPR 2023.

**Questions:**

-The number of Fd_i encoded by the authors is related to the number of source frames. From Fig. 3, it seems that during the GRU update process, they are directly concatenated, which will cause the channel number to be related to the number of source frames. So, if the model uses 5 frames in training and then the number of frames cannot be changed during testing, is that correct?

-From the numerical results and method description, the paper has completely abandoned the use of depth priors now. But if the depth prior is known and the prior is more compact than the depth range calculated in Sec 3.2, can using the depth prior improve the performance of this method?

-As a GRU-type method, how does the accuracy of the estimated results change with the number of iterations?

**Limitations:**

-Limited performance: From the experimental results, the method in this paper is not outstanding on the benchmark, and the gap with the state-of-the-art methods is quite obvious, even only compared with GRU-based methods (such as CER-MVS).

-Limited novelty: The method in this paper is more like a combination of existing modules.

---

> ### Author Rebuttal · Authors · 2024-08-06
>
> > the left side of Eq. 5 should be $V_i(p0)$; The $Fd_i$ in Fig. 2, Eq. 8, and $F^d_i$ in Sec 3.4, the $H_i$ in Fig. 2 and $Hd_i$ in Eq. 8.
>
> Thanks for your  reminder. I will standardize the notation for cost volume as  $V_i(p0)$, epipolar disparity features as $Fd_i$, and disparity hidden State as $Hd_i$ to eliminate the gap between the formulas and the figures.
>
> > the use of symbols is not standardized, such as the representation of matrices and vectors.
>
> We will recheck the entire document for correct notation usage, including matrix and vector parts, and make the necessary corrections.
>
> > In Eq. 8, there is an extra dot multiplication symbol before $\varphi_4$.
>
> Additionally, we will remove the redundant dot multiplication symbol before $\varphi_4$ in Eq. 8.
>
> > For the final GRU update part, the paper lacks a complete description.
>
> We follow the structure of DispMVS, iteratively updating the epipolar flow for each source image. In each iteration, the input to the update operator includes the hidden state, the disparity feature $Fd_i$ output from MDA module, the current epipolar flow, and the context feature of the reference image. The output of the update operator includes a new hidden state, an increment to the disparity flow, and the weight.
> We obtain the depth from the disparity flow and utilize a weighted sum for the depth in a multi-view situation. After fusion, the depth is converted back to disparity flow to perform the next iteration.
>
> > There is an error in the citation of DispMVS[5]
>
> Thank you very much for your reminder. We will correct the citation accordingly.
>
> > Cross-attention is only calculated at the same pixel positions in different frames, making this design very illogical.
>
> It was misunderstood that the attention is not on the sample pixels' image features in different frames but on the disparity features encoded from the cost volume. This cost volume is obtained by back-projecting the sampled depths into the source images. Therefore, the features among multiple frames are associated through the sampled points' depths. Additionally, we enhance the implicit disparity relationships among multi-view frames by encoding pose embedding, which introduces multi-view relative pose information and geometric information between specific sampled points.
>
> > Missing comparisons: CER-MVS, MVSFormer series, GeoMVSNet. \& Limited performance
>
> Thank you for your suggestion. We include CER-MVS, GeoMVSNet, and MVSFormer++ in our comparative experiments. However, these methods all require the depth prior, whereas our algorithm is designed to operate in a depth-range-free setting.
> In paper, we mainly focus on experiments related to Depth Range in Sec. 4.5. And our main comparison is with depth-range-free methods like DispMVS, which reduce dependence on depth priors through network design.
> For methods that rely on depth range priors, whether based on GRU or Transformer, they may exhibit better performance with accurate depth priors. However, their performance significantly degrades when there is an error in the depth prior.
> Due to the length limitations, the detailed explanation and experiments for this part are placed in the Author Rebuttal.
>
>
> > There is a lack of separate ablation experiments on uncertainty and disparity feature hidden states.
>
> Due to the length limitations of the rebuttal, the detailed explanation and experiments for this part are placed in the Author Rebuttal.
>
> > During the GRU update process, the channel number to be related to the number of source frames, and the number of frames cannot be changed during testing?
>
> This is not the case. In Fig. 3, "concat" refers to concatenating the disparity features output from self-attention and cross-attention. After the MDA module, similar to DispMVS, the disparity feature corresponding to each source image is fed to GRU updating module individually. However, since we have already performed extensive global information interaction, the epipolar flows obtained by multi-view situation will be interconnected.
>
> > If the depth prior is known and the prior is more compact than the depth range calculated in Sec 3.2, can improve the performance?
>
> Due to the length limitations of the rebuttal, the detailed explanation and experiments for this part are placed in the Author Rebuttal.
>
> > As a GRU-type method, how does the accuracy of the estimated results change with the number of iterations?
>
> As shown in Fig. 3, the depth error decreases progressively with each iteration.
> The vertical axis represents the depth error, and the horizontal axis represents the number of iterations. Iterations 0-7 correspond to the coarse stage, while iterations 8-9 correspond to the fine stage.
> Fig. 3  shows the depth maps on DTU, in which we can see that the depth map recovers from coarse to fine.
>
> > Limited novelty: The method in this paper is more like a combination of existing modules.
>
> Driven by practical application needs, we creatively proposed applying transformer operations to the disparity features constructed after 2D sampling to remove the dependency on depth range. Additionally, we were the first to address the issue of depth mismatches among different source images during 2D sampling. By designing uncertainty and pose embedding, we endowed the features with geometric relationships, making multi-frame consistent estimation more efficient and accurate.

---

> > ### Comment · Reviewer_Q3xc · 2024-08-13
> > **About the limited novelty**
> >
> > I appreciate for the authors' response. Now, I think I understand this work. It is heavily based on DispMVS, but there is a main improvement.
> >
> > 1. Replace the vanilla feature correlation (named SIMI) in DispMVS with Disparity Feature Encoding combining Multi-view Disparity Attention
> >
> > Besides, there are several less important improvement
> >
> > 1. The variance-based uncertainty, which replaces the sum weight in original DispMVS and is also used in disparity feature encoding.
> > 2. The initialization method for each flow map
> >
> > It shows effectiveness of the proposed modules, although being with more compuration costs.
> >
> > I still have concerns about the model performance (especially when being compared with CERMVS, which is also a GRU based method), and therefore I decide to keep my rating

---

> ### Author Response · Authors · 2024-08-08
> **Separate ablation experiments**
>
> We sincerely apologize for not noticing the failure in transferring the ablation experiments due to a system refresh.
>
> > There is a lack of separate ablation experiments on uncertainty and disparity feature hidden states.
>
> We add the separate ablation experiments. Compared to the performance without the MDA module, this demonstrates the effectiveness of the uncertainty and disparity feature hidden state update modules.
>
> |                                    |   Acc.(mm)↓   |   Comp.(mm)↓   |  Overall(mm)↓  |
> | :--------------------------------- | :-------------: | :-------------: | :-------------: |
> | No Disparity Feature Hidden States |      0.370      |      0.354      |      0.362      |
> | No Uncertainty                     |      **0.336**      |      0.378      |      0.357      |
> | Ours                               | 0.338 | **0.331** | **0.335** |

---

> ### Author Response · Authors · 2024-08-13
> **About the model performance**
>
> Thanks for your feedback. Regarding the performance comparison test of CER-MVS with respect to depth gt priors, we have included this in the Author Rebuttal. We hope this response can address your concerns.

---

> ### Author Response · Authors · 2024-08-14
> **Response to Reviewer**
>
> Dear Reviewer:
>
> We are pleased that our response addressed your questions. We notice that your rating is still borderline reject, and we sincerely want to know if there are any remaining concerns about this work.

---

> > ### Author Response · Authors · 2024-08-14
> > **Response to Reviewer**
> >
> > Thanks very much for your recognition of our work.

---

### Official Review · Reviewer_fCJ7 · 2024-07-13

**Soundness:** 3
**Presentation:** 3
**Contribution:** 3
**Rating:** 5
**Confidence:** 4

**Summary:**

- The author proposes a depth-range-free multi-view stereo framework that simultaneously takes into account all the source images.
- The author has specially designed a 3D pose embedding to better encode specific geometric information.

- The Multi-View Stereo method proposed in the paper achieves more robust depth estimation by cascading Disparity Feature Encoding, Multi-view Disparity Attention, and GRU Updating, and has demonstrated good performance across multiple datasets.

**Strengths:**

- The depth-range has always been an issue affecting the robustness of Multi-View Stereo algorithms. Although previous methods have attempted to address it, they are still imperfect. The method proposed by the author exhibits technical novelty.

- The iterative updating approach proposed by the author can simultaneously take into account the information of all source images.

- The method presented in the article has shown good performance on the dataset's benchmark.

**Weaknesses:**

- The visual results of the method proposed in the article exhibit many floater artifacts.

- The overall description of the network architecture needs further refinement. The writing need to be improved. The current introduction is not well-balanced in detail, and there are several typographical errors with symbols.

- The use of symbols like $F^d_i$ is inconsistent in terms of superscripts and subscripts, making it difficult to understand and appearing unprofessional in typesetting. This is not the only symbol with usage issues; multiple symbols have similar problems. It is recommended to provide a comprehensive explanation of the symbols in the supplementary material and to revise the symbols used in the figures and text.

**Questions:**

- Are the training results of the epipolar disparity flows truly meaningful? Are there any visual results to demonstrate this?

- How is $D_0$ constructed? During construction, were all source images initialized with the reference image? Was $D_0$ calculated as an average of initial values obtained from each source view? The explanation could be clearer.

- How is the depth updated through the GRU? The section 3.5 is not clearly explained.

- The ablation study is not sufficient:
  - What are the different initializations of depth, such as random or all zeros?
  - How is the convergence?

- In Figure 2, what is the relationship between the $e_i$ output from the GRU and the $e_i$ after fusion? How is $D_i$ obtained from $e_i$, and how is $U_i$ combined?

- As a method that also uses GRU for updates, there is a lack of reference to and discussion of "Multiview Stereo with Cascaded Epipolar RAFT."

**Limitations:**

The author has already addressed the limitations of the method.

---

> ### Author Rebuttal · Authors · 2024-08-06
>
> > The visual results of the method proposed in the article exhibit many floater artifacts.
>
> Thanks for your reminder. Artifacts are generated during the depth fusion step. For point cloud fusion, we directly sampled the pcd method from DispMVS. This method selects multiple relevant depth views for each image to perform back-projection. After threshold filtering, the back-projected depth is weighted and combined with the current depth, which can lead to the generation of artifacts.
>
> > The overall description of the network architecture needs further refinement. The writing need to be improved. The current introduction is not well-balanced in detail, and there are several typographical errors with symbols.
>
> We will thoroughly review the entire manuscript to complete the details of the network design and correct any inconsistencies between the formulas and symbols in the figures.
>
> > The use of symbols like $F_i^d$ is inconsistent in terms of superscripts and subscripts. It is recommended to provide a comprehensive explanation of the symbols in the supplementary material and to revise the symbols used in the figures and text.
>
> Thanks for your  reminder. I will standardize the notation for epipolar disparity features as $Fd_i$. I will carefully align the symbols in the formulas and images throughout the entire manuscript.
>
> > Are the training results of the epipolar disparity flows truly meaningful? Are there any visual results to demonstrate this?
>
> To consider the information interaction of multiple source images during 2D sampling, it is necessary to train the flow. As shown in Fig. 1, after the MDA and GRU modules, the flow obtained from different source images is consistent in terms of details. By incorporating geometric information, although the flow magnitudes on different source images vary, the representation of edges and other details is unified.
>
> > How is $D_0$ constructed? During construction, were all source images initialized with the reference image?
>
> On L.151, we describe that $D_0$ is obtained by selecting the midpoint of the sampling range. On L.153, we first initialize $D_0$ for all source images and then take the average as the initial depth obtained from each source view.
> Thank you for your suggestion, we will make this description clearer.
>
> > How is the depth updated through the GRU?
>
> We follow the structure of DispMVS, iteratively updating the epipolar flow for each source image. We obtain the depth from the disparity flow and utilize a weighted sum for the depth in a multi-view situation. After fusion, the depth is converted back to disparity flow to perform the next iteration.
>
> > The use of symbols like $F_i^d$ is inconsistent in terms of superscripts and subscripts, symbols' problems.
>
> Thanks for your  reminder. We will standardize the notation for cost volume as $V_i(p0)$, epipolar disparity features as $Fd_i$, and disparity hidden State as $Hd_i$ to eliminate the gap between the formulas and the figures.
> We will recheck the entire document for correct notation usage, including matrix and vector parts, and align the symbols in the formulas and images.
>
> > What are the different initializations of depth, such as random or all zeros? How is the convergence?
>
> We add the comparison with different initializations of depth. However, due to the unknown depth range, it is not feasible to design a random sampling range for 3D sampling, which is impractical. Additionally, when the initial depth is set to 0, significant noise can occur during feature warping. Therefore, we designed three sets of ablation experiments: randomly initializing within the epipolar search range, and initializing fixed at the left endpoint of the epipolar line.
>
> |                    |   Acc.(mm)↓   |   Comp.(mm)↓   |  Overall(mm)↓  |
> | :----------------- | :-------------: | :-------------: | :-------------: |
> | The left endpoint  |      0.389      |      4.373      |      2.381      |
> | randomly 2D sample |      2.597      |      20211      |      2.404      |
> | Ours               | **0.338** | **0.331** | **0.335** |
>
> For each case, as the GRU iteratively updates, the error gradually decreases, and the depth gradually converges. Fig. 4. shows the convergence behavior when the initial point is set to the left endpoint.
>
> > In Figure 2, what is the relationship between the $e_i$ output from the GRU and the $e_i$ after fusion? How is $D_i$ obtained from $e_i$, and how is $U_i$ combined?
>
> In each iteration, the GRU update operator output the current epipolar flow $e_i$ and weight $U_i$. We obtain the depth from the disparity flow by Eq.3 and utilize a weighted sum for the depth in a multi-view situation. After fusion, the depth is converted back to disparity flow $e_i$ to perform the next iteration.
>
> > As a method that also uses GRU for updates, there is a lack of reference to and discussion of "Multiview Stereo with Cascaded Epipolar RAFT.
>
> Please read the detailed explanation in the Author Rebuttal section for this part of the response.

---

> > ### Comment · Reviewer_fCJ7 · 2024-08-13
> >
> > I appreciate the authors' positive response and their clarification of my questions, as well as their commitment to revising symbols and expressions. However, I still have a few concerns.
> >
> > Firstly, I am unable to understand why there are many floater artifacts in the DTU data. The authors' response was somewhat vague, although I believe these issues could be addressed by carefully tuning the depth fusion parameters.
> >
> > Additionally, the use of symbols by the authors seems confusing. For example, using multiple letters to represent a single variable, such as $Fd_i$ instead of $F^d_i$, appears unconventional.
> >
> > Moreover, I think the authors should include a more comprehensive discussion comparing their method with CER-MVS, especially regarding the use of depth range, as pointed out by Reviewer Q3xc, which currently seems inaccurate.
> >
> > Overall, I am inclined to adjust the score to Borderline Accept.

---

> ### Author Response · Authors · 2024-08-13
> **Response to concerns**
>
> Thanks for your reply and positive feedback. In response to your concerns, we provide the following answers:
>
> > I am unable to understand why there are many floater artifacts in the DTU data.
>
> Thanks for your reminder. As shown in Fig. 2 of PDF in Author Rebuttal, we compare our method with MVSFORMER++(SOTA) and find that point clouds inevitably exhibit artifacts in current MVS methods including the state-of-the-art methods. Artifacts are generated during the depth fusion step. For point cloud fusion, the pcd method selects multiple relevant depth views for each image to perform back-projection. After threshold filtering, the back-projected depth is weighted and combined with the current depth, which can lead to the generation of floater artifacts. This is due to 2D depth errors and inconsistencies across multi-view frames. Adjusting the threshold can mitigate this issue but may affect the overall quality of the 3D point cloud.
>
> > The use of symbols by the authors seems confusing.
>
> Thanks for your suggestion, we will make the adjustments accordingly.
>
> > more comprehensive discussion comparing their method with CER-MVS.
>
> CER-MVS uses depth gt priors to calculate the scale of scene and the way varies across different datasets. CER-MVS primarily uses three datasets: BlendedMVS dataset, Tanks-and-Temples dataset, and DTU dataset. BlendedMVS dataloader (https://github.com/princeton-vl/CER-MVS/blob/main/datasets/blended.py) provides two scaling methods: the default method (self.scaling == "median") uses depth gt to scale the scene to a median of 600 mm on Line 72, while the alternative method scales the scene using the depth range prior provided by the dataset (labeled as 'depth range gt') to achieve a minimum of 400 mm on Line 75. Tanks-and-Temples dataloader (https://github.com/princeton-vl/CER-MVS/blob/main/datasets/tnt.py) uses depth range prior to scale the scene to achieve a minimum of 400 mm. In the code, these depth range gt priors are loaded and represented by the 'scale_info' variable on Line 74 and 75. For DTU dataloader (https://github.com/princeton-vl/CER-MVS/blob/main/datasets/dtu.py), the depth range gt is not directly loaded in dataloader because DTU dataset already has a depth median of 600 mm and a minimum depth of 400 mm. The scene scale meets the network's requirements.
>
> CER-MVS performs uniform sampling on inverse depth, fixing the depth sample range (https://github.com/princeton-vl/CER-MVS/blob/main/core/corr.py). In the code (https://github.com/princeton-vl/CER-MVS/blob/main/core/raft.py), the maximum disparity $d_{max}$ is set to 0.0025, and the disparity increments of stage1 and stage2 are set to $d_{max}/64$ and $d_{max}/320$ on Line 81. By scaling the dataset, CER-MVS can obtain more accurate depth increments and maintain updates within the inverse depth sampling interval.
>
> To validate the robustness of CER-MVS to depth gt priors, we have designed experiments by altering the median or minimum depth values of the scene. Specifically, we have introduced noise perturbations to change the median of DTU dataset during rebuttal.
>
> When there is no per-pixel depth gt, CER-MVS uses the depth range to scale the scene depth to a minimum value of 400 mm. It is worth noting that the depth range provided by the dataset is very accurate. For instance, the ground-truth data for Tanks-and-Temples dataset is captured using an industrial laser scanner. However, in practical applications, the depth range obtained through COLMAP are often inaccurate due to the sparsity of feature points and issues such as occlusion and suboptimal viewpoint selection. To verify the robustness of depth range priors, we use the depth range obtained from COLMAP to replace the depth range gt.
>
> | Depth Range | Method |   Acc.(mm)↓   |   Comp.(mm)↓   |  Overall(mm)↓  |
> | :---------: | :-----: | :-------------: | :-------------: | :-------------: |
> |     GT     | CER-MVS |      0.359      | **0.305** | **0.332** |
> |     GT     |  Ours  | **0.338** |      0.331      |      0.335      |
> |   COLMAP   | CER-MVS |      0.816      | **0.326** |      0.571      |
> |   COLMAP   |  Ours  | **0.338** |      0.331      | **0.335** |
>
> From the table, it can be seen that CER-MVS exhibites a certain degree of decline due to the noise in the depth range caused by COLMAP. In contrast, our method, which is independent of depth range, maintained consistent performance regardless of changes in depth range. This further demonstrates the necessity of eliminating the depth priors.

---

> > ### Comment · Reviewer_fCJ7 · 2024-08-14
> >
> > Thank you for the response and clarification. I believe the current further response has addressed some of my concerns. I will give Borderline Accept. I believe that if the revised version of the paper addresses these points and incorporates updates, its overall quality would range between Borderline Accept and Weak Accept.

---

> > > ### Author Response · Authors · 2024-08-14
> > > **Response to Reviewer**
> > >
> > > Thanks for your positive feedback on our work. We promise to address the corresponding points in the final version.

---

### Official Review · Reviewer_heJr · 2024-07-14

**Soundness:** 3
**Presentation:** 3
**Contribution:** 3
**Rating:** 5
**Confidence:** 4

**Summary:**

The paper proposes a depth-rage free method for MVS. It leverages the transformers for designing a global-aware model, with using pose positional embedding to guide the model and also predict the uncertainty at the same time. The methods demonstrates good performance on diverse datasets and is also robust to different depth range.

**Strengths:**

- The idea of building global-aware models using transformers makes sense.
- I like the idea of injecting inductive bias using pose positional embedding and the uncertainty method.
- The proposed method demonstrates satisfying experimental results and robustness to different depth range.

**Weaknesses:**

- Related work. Using transformers for building global-aware model is very common for 3D reconstruction now [1,2,3]. I would be good if the authors could discuss these related works.

[1] Wang, Peng et al. “PF-LRM: Pose-Free Large Reconstruction Model for Joint Pose and Shape Prediction.” ICLR 2024. [2] Jiang, Hanwen et al. “LEAP: Liberate Sparse-view 3D Modeling from Camera Poses.” ICLR 2024. [3] Zhang, Kai et al. “GS-LRM: Large Reconstruction Model for 3D Gaussian Splatting.” ArXiv 2024.

- Dense view inputs. The experiments are performed in the sparse-view setting where the computation of transformers is not a big problem. Is the method also efficient in the dense view setting? If not, I hope the authors could have more discussions in the paper.

- I would say that the gains in the ablation experiments are not that significant.

**Questions:**

Please see the weakness section.

**Limitations:**

The scale of the experiment is not big enough. It would be great if the authors could perform larger-scale experiments, ideally Dust3r-level scales, to understand the scaling capability of the proposed method. I can understand it is not easy to acquire more resources for training, but it would be great if the authors could have more discussions on this.

---

> ### Author Rebuttal · Authors · 2024-08-06
>
> > Using transformers for building global-aware model is very common for 3D reconstruction now [1,2,3]. I would be good if the authors could discuss these related works.
>
> Thank you for your suggestion. We will discuss the corresponding references in the related work section.
>
> [1] Wang, Peng et al. “PF-LRM: Pose-Free Large Reconstruction Model for Joint Pose and Shape Prediction.” ICLR 2024.
>
> [2] Jiang, Hanwen et al. “LEAP: Liberate Sparse-view 3D Modeling from Camera Poses.” ICLR 2024.
>
> [3] Zhang, Kai et al. “GS-LRM: Large Reconstruction Model for 3D Gaussian Splatting.” ArXiv 2024.
>
> > Is the method also efficient in the dense view setting?
>
> For images with a resolution of 512x640, our method can handle up to 77 views in V100 during testing phase, and up to 5 views simultaneously during training phase. Using the model trained on the DTU dataset with 5 views, we verify the impact of the number of views on system accuracy. Due to DTU providing co-visible relationships for only 10 frames, we selected 10 frames as the upper limit. The results are shown in the following table, which uses 2D metric. It can be observed that the error decreases as the number of used views increases and gradually coverage. Increasing the view during training may further increase the performance after convergence.
>
> |             | 3 views | 4 views | 6 views | 8 views |
> | :---------- | :-----: | :-----: | ------- | ------- |
> | Depth Error |  5.375  |  5.108  | 4.964   | 4.967   |
>
>
> > The gains in the ablation experiments are not that significant.
>
> We respectfully argue that the gain of adding the proposed components is large enough. For example, the performance gain of adding uncertainty is 9.95\% and that of adding pose embedding is 9.46\%. Besides, we also test the gain in the 2D metric (absolute depth error), the gain of adding uncertainty and pose embeddings is 1.46\% and 35.62\%, respectively.
>
> > Perform larger-scale experiments, ideally Dust3r-level scales, to understand the scaling capability of the proposed method.
>
> We collected some real-world data to test our method in more large-scale environment and also test its generalization. The camera intrinsic and camera pose are both obtained by running COLMAP. As shown in Fig. 5, our model is capable of generating dense point cloud reconstructions for the collected data, which shows basic generalization ability. However, the accuracy of these reconstructions is somewhat lacking. We guess one of the primary factors is the inaccurate camera pose from the COLMAP. It is a promising direction to explore the joint optimization of the camera pose for the MVS in the future. Besides, the limited training data also hinders the performance of our method. Compared with Dust3r which is trained with a mixture of eight datasets, covering millions of image data, our method and other MVS methods are only trained on DTU or BlendedMVS. How to utilize these large datasets for training MVS is also one of our future work.

---

### Official Review · Reviewer_RLMs · 2024-07-15

**Soundness:** 3
**Presentation:** 1
**Contribution:** 3
**Rating:** 5
**Confidence:** 4

**Summary:**

The paper describes an MVS approach that does not depend on given depth ranges, which MVS algorithms typically require when building 3D cost volumes. The solution, previously proposed in DispMVS, is to perform searching and iterative updates in disparity space. Compared to DispMVS, the authors propose several novel designs to further eliminate the dependency on depth range during initialization, as well as to allow synchronous updates from all source views.

**Strengths:**

**Significance**: The complete elimination of the dependency on depth ranges brings convenience and avoids performance degradation due to improper ranges. These benefits may enable wider adoption of MVS in real-world applications.

**Originality**: Although the paper largely follows the foundation built in DispMVS, the several improvements are novel and nontrivial. The combination of 3D pose embedding and cross-attention as a solution for synchronous updates among all source views is quite interesting and proven effective.

**Quality**: The results show consistent improvements over other depth-range-free options. The ablation studies and the analysis of impact from varying depth ranges help make the overall design more convincing.

**Weaknesses:**

**Clarity**
The paper has some rather noteworthy issues regarding writing. The method section needs to have the right level of detail. The approach is built on top of DispMVS and shares many concepts and details, and the authors' choices to keep and omit details seem somewhat arbitrary, resulting in a non-self-contained method section. For example, there are also some concepts used without introduction, e.g. "sampling range" on L.168, "t_c, t_f" in Eq.11. The GRU updating process isn't defined. The relationship between epipolar disparity flow and depth is a critical concept that the paper does not explain.

The resulting poor readability is potentially a significant issue, but unfortunately, it is hard to address in a rebuttal. I'd like to see how other reviewers weigh its impact.

**Performance**
While beating other depth-range-free solutions, including DispMVS, on the two benchmarks, results still need to catch up on cost-volume approaches like GBi-Net, often by a large margin. It'll be quite valuable to understand what causes such discrepancy. Is it due to the limited receptive field, search gratuity, or else? Can improving on any of these help close the gap?

**Generalization**
There is no discussion of generalization capability in the paper. Conceptually, it seems a depth-range-free approach would be more generalizable due to being invariant to object/scene scales. The ability to train a single model that operates in various use cases (indoor, outdoor, etc.) is arguably the most desirable feature of such approaches.

**Questions:**

I look forward to answers to the questions above regarding clarity, generalization and performance.

**Limitations:**

The authors recognize speed as a limitation. The performance gap behind cost-volume counterparts is also worth mentioning.

---

> ### Author Rebuttal · Authors · 2024-08-06
>
> > "sampling range" on L.168.
>
> Thanks for your reminder. Following DispMVS, the "sampling range" on L.168 refers to the set of sampling points uniformly sampled along the epipolar line.
>
> > "$t_c$, $t_f$" in Eq.11.
>
> on L.163, "$t_c$, $t_f$" are iterations at the coarse and fine stage.
>
> > The GRU updating process isn't defined.
>
> For GRU updating process, the epipolar flow is iteratively updated for each source image. In each iteration, the input to the update operator includes the hidden state, the disparity feature output from MDA module, the current epipolar flow, and the context feature of the reference image. The output of the update operator includes a new hidden state, an increment to the disparity flow, and the weight. We get the depth from disparity flow and utilize a weighted sum to the depth from multi-view situation. After fusion, the depth is converted back to disparity flow to perform the next iteration.
>
> > The relationship between epipolar disparity flow and depth is a critical concept that the paper does not explain.
>
> For the relationship between epipolar disparity flow $e_i$ and depth $d_0$, we can obtain the position $p_i$ in the source image by adding epipolar disparity flow $e_i$ to initial position $p_i^0$, then we can get $d_0$ by triangulation as Eq.3. Similarly, after obtaining $d_0$, we get the position $p_i$ by Eq.2, and subtract the initial position $p_i^0$ to obtain $e_i$. We will clarify this relationship more clearly in paper.
>
> > Results still need to catch up on cost-volume approaches like GBi-Net, often by a large margin. Is it due to the limited receptive field, search gratuity, or else? Can improving on any of these help close the gap?
>
> Compared to other methods that uniformly sample depth based on a depth range, our network requires a more powerful retrieval capability to regress the correct depth due to the lack of depth range priors and outperform known depth-free methods. There are two main reasons for the discrepancy between our method and cost-volume approaches.
>
> One reason is the search gratuity. Although our method addresses the receptive field problem to a large extent by sampling points along the epipolar line at features with different scales, iterating over the depth range of $(0,\infty)$ significantly increases the search gratuity compared to uniformly sampling within a predefined depth range.
>
> The second reason is the development potential of datasets for deep learning network. Existing MVS datasets, such as DTU, have a relatively uniform depth distribution, mostly around a mean depth of 600. This allows methods directly based on a narrow predefined depth range to achieve precise convergence, limiting the advantage of our method. However, in real-world scenarios, there are many scenes with a wide depth distribution, such as near and far objects, where the background cannot be crudely masked out like in the DTU dataset. In such cases, depth cannot be recovered with a narrow depth prior, necessitating an expanded depth search range, which increases the difficulty of convergence inevitably. I think enhancing accuracy over a large depth range is a crucial problem that MVS must address in the future. Currently, our ongoing work focuses on optimizing the acquisition of initial values and constructing more diverse datasets to endow the network with stronger learning capabilities.
>
> > There is no discussion of generalization capability in the paper.
>
> To evaluate the generalization of our method, we used an iPad to capture data and add inference experiments in multiple scenes. We collected images from various real-world environments, with intrinsic and extrinsic parameters obtained by running COLMAP. As shown in Fig. 5, our model is capable of generating dense point cloud reconstructions for both indoor and outdoor environments. However, the accuracy of these reconstructions is somewhat lacking, which is affected by the following three factors:
>
>
> Inaccurate Camera Pose: The pose estimation from COLMAP is far from satisfying. In contrast, the evaluation benchmark provides more accurate camera pose. For example, DTU uses a structured light scanner and MATLAB calibration toolbox for camera pose estimation. The inaccurate camera pose can lead to large error during the MVS process.
>
> Image Quality Issues: As illustrated in Fig. 6, issues such as overexposed, inadequate lighting and blurriness affect the reconstruction quality. These deficiencies in image quality contribute to the observed inaccuracies in the point cloud.
>
>
> Training Data Limitations: Our model was trained on the DTU dataset, which is relatively small and features a narrow range of scenes. While our model can effectively mitigate depth range effects in various environments, it still struggles with fine detail accuracy. The limited diversity of the DTU dataset constrains the model's ability to capture detailed features accurately. Constructing an MVS dataset with diverse scenes is a promising approach to enhance the robustness and accuracy of point cloud reconstructions.

---

> > ### Comment · Reviewer_RLMs · 2024-08-13
> >
> > I appreciate the detailed responses. The clarifications are useful, though it remains to be seen how well they will be integrated into a final draft. The comments regarding generalization and gaps in performance are reasonable; however, they did not directly address the concerns. Given the significant practical advantages afforded by a range-free model, I'm still keeping a borderline recommendation.

---

### Author Rebuttal · Authors · 2024-08-07

We sincerely thank all reviewers for valuable feedback and positive comments like "novel and nontrivial"(Reviewer RLMs), "satisfying experimental results"(Reviewer heJr), "exhibits technical novelty"(Reviewer fCJ7), "motivation for the model design is clear"(Reviewer Q3xc), "promising results ... to different depth ranges"(Reviewer s1Ui). We will correct all typos in the final version. We will release the code to facilitate more practical research on depth-range-free methods upon acceptance.

The following are some detailed answers.

>Add comparisons: CER-MVS, MVSFormer++, GeoMVSNet

We include CER-MVS, GeoMVSNet, and MVSFormer++ in our comparative experiments. However, these methods all require the depth range prior, whereas our algorithm is designed to operate in a depth-range-free setting.

In paper, we mainly focus on experiments related to Depth Range in Sec. 4.5. And our main comparison is with depth-range-free methods like DispMVS, which reduce dependence on depth priors through network design. For methods that rely on depth range priors, whether based on GRU or Transformer, they may exhibit better performance with accurate depth priors.
However, their performance significantly degrades when there is an error in the depth prior.

|   Depth Range   |   Method   |   Acc.(mm)↓   |   Comp.(mm)↓   |  Overall(mm)↓  |
| :-------------: | :---------: | :-------------: | :-------------: | :-------------: |
|  （425，935）  | MVSFORMER++[Ⅱ] | **0.309** | **0.252** | **0.281** |
|  （425，935）  |   GeoMVS[Ⅲ]   |      0.331      |      0.259      |      0.295      |
|  （425，935）  |    Ours    |      0.338      |      0.331      |      0.335      |
| （212.5，1870） | MVSFORMER++[Ⅱ] |      0.361      | **0.319** |      0.340      |
| （212.5，1870） |   GeoMVS[Ⅲ]   |      0.374      |      0.415      |      0.394      |
| （212.5，1870） |    Ours    | **0.338** |      0.331      | **0.335** |
| （141.6，2805） | MVSFORMER++[Ⅱ] |      0.739      |      0.820      |      0.780      |
| （141.6，2805） |   GeoMVS[Ⅲ]   |      0.602      |      1.663      |      1.133      |
| （141.6，2805） |    Ours    | **0.338** |      **0.331**      |      **0.335**      |

CER-MVS does not directly sample within the depth range but requires the use of depth ground truth (depth gt) to scale the entire scene to a range with a mean of 600. When we introduce a certain amount of noise to the depth gt, thereby altering the scale, the performance of CER-MVS declines sharply.

| Depth GT Noise | Method |   Acc.(mm)↓   |   Comp.(mm)↓   |  Overall(mm)↓  |
| :------------: | :-----: | :-------------: | :-------------: | :-------------: |
|       0       | CER-MVS[Ⅰ] | 0.359 | **0.305** | **0.332** |
|      20\%      | CER-MVS[Ⅰ] |      9.230          |       10.236         |       9.858         |
|      30 \%     | CER-MVS[Ⅰ] |      9.385      |     10.098     |      9.741      |
|                |  Ours  | **0.338** |      **0.331**      |      **0.335**      |

We also add CER-MVS, GeoMVSNet, and MVSFormer++ to depth range experiment in Appendix B.4. Experiments demonstrate that for methods dependent on depth range prior, even if a rough depth range can be obtained from COLMAP, their performance still degrades.


| Depth Range |   Method   | Acc.(mm)↓ | Comp.(mm)↓ | Overall(mm)↓ |
| :---------: | :---------: | :--------: | :---------: | :-----------: |
|     GT     | MVSFORMER++[Ⅱ] | **0.309** | **0.252** |  **0.281**  |
|     GT     |   GeoMVS[Ⅲ]   |   0.331   |    0.259    |     0.295     |
|     GT     |    Ours    |   0.338   |    0.331    |     0.335     |
|   COLMAP   | MVSFORMER++[Ⅱ] |   0.361   | **0.319** |     0.340     |
|   COLMAP   |   GeoMVS[Ⅲ]  |   0.374   |    0.415    |     0.394     |
|   COLMAP   |    Ours    | **0.338** |    0.331    |  **0.335**  |


> If the depth prior is known and the prior is more compact than the depth range calculated in Sec 3.2, can improve the performance?

To validate the effectiveness of the depth prior, we designed the experiment. The initial depth is reverted to random initialization based on the depth prior, similar to DispMVS. The results are shown as follows. It can be observed that adding depth prior can improve performance to some extent, but the difference compared to our depth-range-free method is not significant. This indicates that the current initial point selection strategy and the design of the Transformer enable the network to regress to the correct range, resulting in accurate depth estimation.

|                                               |   Acc.(mm)↓   |   Comp.(mm)↓   |  Overall(mm)↓  |
| :-------------------------------------------- | :-------------: | :-------------: | :-------------: |
| Random Depth Initialization among Depth Range |      **0.331**      |      0.335      |      **0.333**      |
| Ours                                          | 0.338 | **0.331** | 0.335 |




[Ⅰ] Ma, Zeyu, Zachary Teed, and Jia Deng. "Multiview stereo with cascaded epipolar raft." ECCV 2022.

[Ⅱ] Cao, Chenjie, Xinlin Ren, and Yanwei Fu. "MVSFormer++: Revealing the Devil in Transformer's Details for Multi-View Stereo." ICLR 2024.

[Ⅲ] Zhang, et al. "Geomvsnet: Learning multi-view stereo with geometry perception." CVPR 2023.

---

> ### Comment · Area_Chair_gxoS · 2024-08-12
> **Rebuttal**
>
> Dear Reviewers,
>
> please provide feedback about the authors' rebuttal; the deadline is approaching.
>
> Thnak you

---

> ### Comment · Reviewer_Q3xc · 2024-08-13
> **The claim and comparison with CER-MVS is uncrediable**
>
> The authors claim that
> >CER-MVS does not directly sample within the depth range but requires the use of depth ground truth (depth gt) to scale the entire scene to a range with a mean of 600. When we introduce a certain amount of noise to the depth gt, thereby altering the scale, the performance of CER-MVS declines sharply.
>
> However, in CERMVS paper, it is claimed that
>
> >To pair neighbor views with reference views, we use the same method as MVSNet. In BlendedMVS, **which is used for training only**, the scenes have large variations in the range of depth values, we scale each reference view, along with its neighbor views, so that its ground-truth depth has a median value **600 mm**. When we evaluate on Tanks-and-Temples, due to lack of ground-truth and noisy background, **we scale each reference view, along with its neighbor views**, so that its minimum depth of a set of reliable feature points (computed by COLMAP as in MVSNet) is 400 mm. To stitch the predicted depth maps from multiple reference views, we simply scale back each depth map to its original scale.
>
> For evaluation and test, CERMVS do not rely on the ground truth. When I check the source code of CERMVS again, there is even no depth ground truth loaded in the test dataloader. Why dose the author claim CERMVS rely on depth gt during inference?

---

> ### Author Response · Authors · 2024-08-13
> **Comparison with CER-MVS**
>
> We respectfully argue that there are some misunderstandings in your interpretation of the scaling operations in the code. CER-MVS uses depth gt priors to calculate the scale of scene and the way varies across different datasets.
>
> CER-MVS primarily uses three datasets: BlendedMVS dataset, Tanks-and-Temples dataset, and DTU dataset. BlendedMVS dataloader (https://github.com/princeton-vl/CER-MVS/blob/main/datasets/blended.py) provides two scaling methods: the default method (self.scaling == "median") uses depth gt to scale the scene to a median of 600 mm on Line 72, while the alternative method scales the scene using the depth range prior provided by the dataset (labeled as 'depth range gt') to achieve a minimum of 400 mm on Line 75. Tanks-and-Temples dataloader (https://github.com/princeton-vl/CER-MVS/blob/main/datasets/tnt.py) uses depth range gt prior to scale the scene to achieve a minimum of 400 mm. In the code, these depth range gt priors are loaded and represented by the 'scale_info' variable on Line 74 and 75. For DTU dataloader (https://github.com/princeton-vl/CER-MVS/blob/main/datasets/dtu.py), the depth range gt is not directly loaded in dataloader because DTU dataset already has a depth median of 600 mm and a minimum depth of 400 mm. The scene scale meets the network's requirements.
>
> CER-MVS performs uniform sampling on inverse depth, fixing the depth sample range (https://github.com/princeton-vl/CER-MVS/blob/main/core/corr.py). In the code (https://github.com/princeton-vl/CER-MVS/blob/main/core/raft.py), the maximum disparity $d_{max}$ is set to 0.0025, and the disparity increments of stage1 and stage2 are set to $d_{max}/64$ and $d_{max}/320$ on Line 81. By scaling the dataset, CER-MVS can obtain more accurate depth increments and maintain updates within the inverse depth sampling interval.
>
> To validate the robustness of CER-MVS to depth gt priors, we have designed experiments by altering the median or minimum depth values of the scene. Specifically, we have introduced noise perturbations to change the median of DTU dataset during rebuttal.
>
> When there is no per-pixel depth gt, CER-MVS uses the depth range to scale the scene depth to a minimum value of 400 mm. It is worth noting that the depth range provided by the dataset is very accurate. For instance, the ground-truth data for Tanks-and-Temples dataset is captured using an industrial laser scanner. However, in practical applications, the depth range obtained through COLMAP are often inaccurate due to the sparsity of feature points and issues such as occlusion and suboptimal viewpoint selection. To verify the robustness of depth range priors, we use the depth range obtained from COLMAP to replace the depth range gt.
>
> | Depth Range | Method |   Acc.(mm)↓   |   Comp.(mm)↓   |  Overall(mm)↓  |
> | :---------: | :-----: | :-------------: | :-------------: | :-------------: |
> |     GT     | CER-MVS |      0.359      | **0.305** | **0.332** |
> |     GT     |  Ours  | **0.338** |      0.331      |      0.335      |
> |   COLMAP   | CER-MVS |      0.816      | **0.326** |      0.571      |
> |   COLMAP   |  Ours  | **0.338** |      0.331      | **0.335** |
>
> From the table, it can be seen that CER-MVS exhibites a certain degree of decline due to the noise in the depth range caused by COLMAP. In contrast, our method, which is independent of depth range, maintained consistent performance regardless of changes in depth range. This further demonstrates the necessity of eliminating the depth priors.

---

> > ### Comment · Reviewer_Q3xc · 2024-08-14
> >
> > Thanks for the response, this address my quetions.

---

### Decision · Program_Chairs · 2024-09-25

**Decision:**

Accept (poster)

**Comment:**

Depth-range agnostic multi-view stereo is an interesting recent research trend, and this paper tackles it by proposing a framework that exploits all source images simultaneously in contrast to competitors in this specific context. On the one hand, the reviewers acknowledged an innovative contribution and good performance, especially compared to depth-range agnostic methods. On the other hand, they highlighted significant issues, mainly regarding clarity, notation and experimental evaluation. The rebuttal/discussion allowed for the clarification of most of these concerns, and finally, all reviewers leaned toward acceptance. Nonetheless, the paper requires a significant revision to increase readability and clarity and integrate all the additional information/experiments that emerged during the discussion with the authors. Additionally, the authors of RAM-Depth provide pre-trained models on Blended (and TartanAir) at https://github.com/andreaconti/ramdepth. Thus, an exhaustive comparison (when training on Blended) with this direct competitor is feasible and required.